# Spatial Effect of Ecological Environmental Factors on Mumps in China during 2014–2018

**DOI:** 10.3390/ijerph192315429

**Published:** 2022-11-22

**Authors:** Li Wen, Danling Yang, Yanning Li, Dongjia Lu, Haixia Su, Mengying Tang, Xiaokun Song

**Affiliations:** 1Department of Biostatistics, School of Public Health, Guangxi Medical University, Nanning 530021, China; 2Guangxi Key Laboratory of AIDS Prevention and Treatment, Guangxi Universities Key Laboratory of Prevention and Control of Highly Prevalent Disease, Nanning 530021, China

**Keywords:** mumps, spatial effect, spatial durbin panel model

## Abstract

(1) Background: although mumps vaccines have been introduced in most countries around the world in recent years, mumps outbreaks have occurred in countries with high vaccination rates. At present, China remains the focus of the global fight against mumps. This study aims to observe the epidemic characteristics and spatial clustering patterns of mumps and to investigate the potential factors affecting the disease incidence, which could provide novel ideas and avenues for future research as well as the prevention and control of mumps. (2) Methods: we used ArcGIS software to visualize the spatial distribution and variation of mumps. Spatial autocorrelation analysis was applied to detect the spatial dependence and clustering patterns of the incidence. We applied the Spatial Durbin Panel Model (SDPM) to explore the spatial associations of ecological environmental factors with mumps. (3) Results: overall, the incidence rate showed a significant upward trend from 2014 to 2018, with the highest number of cases in the 10–15-year age group and from May to June. Geographically, the high incidence clusters were concentrated in southern regions, including Hunan, Hubei, Chongqing, Guizhou, Guangdong, and Guangxi. This study also found that mumps has a positive spatial spillover effect in the study area. The average temperature and GDP of the local and adjacent areas have a significant impact on mumps. The increase in PM2.5 contributes to the rise in the incidence of mumps in this region. (4) Conclusions: these results can offer some novel ideas for policymakers and researchers. Local meteorological conditions and economic levels can extend to surrounding areas to affect the occurrence of mumps, so regional cooperation becomes particularly important. We recommend investment of public health funds in areas with a high incidence of mumps and developing economies to reduce and control the incidence of mumps.

## 1. Introduction

Mumps is a highly contagious disease that often occurs in children and adolescents [1]. Respiratory droplet transmission and close contact are the main routes of disease transmission. Although mumps has a mild onset, it can lead to serious and complex complications, such as orchitis or oophoritis, which can increase the risk of infertility in both women and men [2]. With the availability of the measles, mumps, and rubella (MMR) vaccine, the risk of this disease and its complications has been lowered drastically, which has eased the economic and disease burden on the population [3]. The World Health Organization reported that 123 Member States had introduced the mumps-containing vaccine (Mucv) nationwide by the end of 2021 [4]. Nevertheless, mumps outbreaks still occur in countries with high vaccination rates, such as Australia [5], the United States [6], and Portugal [7]. Outbreaks occur not only in children but also increasingly in adults. China, with its vast territory and complex geography, has become a priority in the global fight against mumps.

With the development of the economy and the expansion of urbanization, the relationship between the environment and disease has become a hot topic for researchers. According to the State of the World’s Air 2020, air pollution led to 6.67 million deaths worldwide in 2019, and particulate matter (PM) is the greatest threat to human health, especially as a risk factor in respiratory diseases [8,9]. A study by Zhang et al. [10] found a significant positive association between PM2.5 and mumps, whereas Xie and his colleagues [11] found no such relationship in Chongqing, and the discrepancy may be related to differences in the scopes of the two studies. So far, the fact that meteorological factors significantly affect mumps is beyond doubt, but the mechanism has not been clarified. Previous studies on mumps and weather variables most commonly used generalized additive models (GAM) and distributed lag nonlinear models (DLNM) and found that meteorological factors, especially air temperature, have a nonlinear relationship with mumps infection risk and demonstrate a lagged effect [12,13,14]. Studies in Guangzhou [12] and Hefei [14] found a higher risk of mumps at extremely high and extremely low temperatures, while the disease risk increased in Taiwan [1] at temperatures above 15 °C and decreased when the temperature was below 29 °C. The studies in Shandong and Guangxi both adopted a two-stage analysis method, which is a combination of time series analysis and multivariate meta-analysis, to explore the relationship between meteorological conditions and diseases in multiple cities and also found different results. Low temperatures were more likely to affect the occurrence of mumps than the high temperatures in Shandong, while the effect of high temperature on mumps was rapid and short, and the effect of low temperature on mumps was slow and long in Guangxi. These findings revealed that discrepancy in the research methods and study area may be a reasonable explanation for the inconsistent results. Worthy of note, the results of a multivariate meta-analysis of the above two articles showed that geographical, socioeconomic, and health service factors were important modifiers of the morbidity of weather-induced mumps [13,15]. The authors suggested that we should raise concern about some economic and social factors, in addition to environmental factors, in the study of mumps. For the reason that there is little research in this field, Fu et al. [16] comprehensively evaluated the association between sociodemographic factors and mumps and found that the proportion of the population aged 0–14 years, the number of health workers, and passengers per capita were positively correlated with the occurrence of mumps. To summarize, meteorological, pollutant and sociodemographic factors are strongly correlated with mumps incidence, but previous studies have not considered whether these factors affect the incidence rate in adjacent areas, which may result in biased estimates. Furthermore, the results of a large number of studies also displayed that the distribution of mumps in space is not random but dependent [17,18]. Therefore, this study identifies the spatial effect of mumps by using a spatial econometric model.

Spatial econometric models have been widely used in tuberculosis, hand, foot and mouth disease (HFMD), AIDS and other infectious diseases [19,20,21,22], but there is still a knowledge gap in the study of mumps. Therefore, this paper aims to use spatial autocorrelation analysis to discover the clustering pattern of mumps and then further detect the spatial association and spillover effect of mumps with meteorological conditions, pollutants, health resources, and economic and social factors by using the Spatial Durbin Panel Model (SDPM) in the spatial econometric models. This provides a novel way for researchers to determine the potential factors and a new clue to mumps control.

## 2. Materials and Methods

### 2.1. Data Source

The reported cases of mumps in mainland China (excluding Taiwan, Hong Kong and Macao due to unavailability of data) from January 2014 to December 2018 were extracted from the Data Center of China Public Health Science established by the Chinese Center for Disease Control and Prevention (CDC) (https://www.phsciencedata.cn/Share/ky_sjml.jsp, accessed on 25 July 2022) [23]. Yearly data on province-level socioeconomic variables and health resource, namely population data, gross domestic product (GDP), the urbanization rate (UR), the number of registered nurses per 1000 capita (RNs), and the proportion of the population educated at the college level and above (EDU) and monthly data on meteorological variables, namely average temperature (TEM), precipitation (PRE) and relative humidity (RH), were all extracted from the National Bureau of Statistics (http://www.stats.gov.cn/tjsj/ndsj/, accessed on 25 July 2022) [24]. Monthly PM2.5 was obtained from the Air Quality Online Monitoring and Analysis Platform (https://www.aqistudy.cn/historydata/, accessed on 25 July 2022) [25]. The above meteorological, pollutant, socio-economic and health resource information are collectively referred to as ecological environmental factors. The incidence rate was calculated with the corresponding formula and expressed as cases per 100,000 population. The vector map file of China was collected from the Resource and Environmental Science and Data Center (http://www.resdc.cn/, accessed on 25 July 2022) [26].

### 2.2. Statistical Analysis

#### 2.2.1. Descriptive Epidemiological Analysis

In this paper, the mumps cases and ecological environmental factors were generally described using a descriptive index (means, standard deviations, and percentiles). The five-year total mumps cases were grouped by age and month to comprehend the epidemiological characteristics of the disease. In addition, the change in mumps in each province from 2014 to 2018 was derived from a time trend graph. Finally, in order to understand the geographical distribution of mumps more intuitively, a visual provincial thematic map was produced.

#### 2.2.2. Three-Dimensional (3D) Trend Surface Analysis

In the 3D trend surface analysis, the least square method was applied to fit a binary polynomial regression model to show the spatial distribution rule and variation trend of mumps [20,27]. The x, y, and z axes of the geometric centre represent the longitude, latitude, and mumps morbidity of the 31 provinces, respectively. In the regression model, the longitude from east to west and the latitude from south to north are independent variables, and the incidence is the dependent variable [21].

#### 2.2.3. Spatial Autocorrelation Analysis

A spatial autocorrelation analysis was performed to determine whether mumps was spatially correlated and, if so, the magnitude of the correlation. The Moran’s index (Moran’s I), a statistic of this analysis, can measure the relationship between the same variables in adjacent spatial units, ranging from −1 to 1 [28]. The index closer to 1 indicates a similar incidence of mumps clustered together (namely clustered), while the index closer to −1 indicates dissimilar incidences clustered together (namely dispersed). When the index is close to 0, the incidence of mumps is randomly distributed, and there is no spatial autocorrelation (namely random distribution) [20]. The global Moran’s index can reflect the spatial dependence of a variable in a whole area, while the local Moran’s I (local indicators of spatial association, LISA) was applied to detect the location and type of the cluster. The spatial patterns of the studied variables in the LISA cluster map were divided into five categories: High-High cluster, Low-Low cluster, Low-High cluster, High-Low cluster and not significant. A High-High cluster represents the spatial relationship of high-incidence regions surrounded by high-incidence regions, and a Low-Low cluster represents the spatial relationship of low-incidence regions surrounded by low-incidence regions [29]. However, the two cluster patterns of high-low and low-high indicate dissimilar morbidity between adjacent areas.

#### 2.2.4. Spatial Econometric Model

Based on the above spatial association of diseases, the spatial econometric model was applied to further determine the influence of ecological environmental variables on mumps. At present, the Spatial Lag Panel Model (SLPM), Spatial Error Panel Model (SEPM), and Spatial Durbin Panel Model (SDPM) are the most common spatial econometric models. The mutual impacts of dependent variables between adjacent regions can be analysed with the SLPM. The SEPM can reflect the spatial dependence caused by the neglected and unobservable variables included in the error term, whereas the SDPM is applied to investigate not only the impacts of local area variables on dependent variables but also those of dependent and independent variables in adjacent regions [30,31]. Considering the possibility of spatial spillover effect in both independent and dependent variables, the SDPM is more suitable for the research purpose of this paper. The natural logarithmic transformation was applied to non-ratio variables to reduce the heteroscedasticity and collinearity of the original data. The SDMP is constructed as follows in this article:(1)$$Incidence_{it}=\propto +\rho_{1}\sum_{i=1}^{n}W_{it}Incidence_{it}+\theta \sum X_{it}+\tau \sum_{i=1}^{n}W_{it}X_{it}+\mu_{i+}\varepsilon_{it}$$

$Incidence_{it}$ is the natural logarithm of the disease incidence in year t in province i, where i ranges from 1 to 31, and t ranges from 1 to 5; $X_{it}$ is ecological environmental variables in year t in province i; α denotes the constant term; $W_{it}$ corresponds to the spatial contiguity weight matrix (rook) of 31 × 31; $\rho_{1}$ and τ are spatial lag coefficients of the variables; θ is the estimation coefficients of ecological environmental variables on mumps; $\mu_{i}$ represents an individual fixed effect; $\varepsilon_{it}$ is the normally distributed random error term [32].

#### 2.2.5. Statistical Software

We calculate variables and make line charts and flow charts (Figure 1) in Microsoft Office 2019 (Microsoft Corp, Redmond, WA, USA). The execution of spatial autocorrelation analysis and the generation of a spatial weight matrix were performed using Geoda version 1.18.0 (the University of Chicago, Chicago, IL, USA). Three-dimensional (3D) trend surface analysis and the visualization of incidence rate distribution and local spatial autocorrelation results were performed in ArcGIS version 10.5 (ESRI, Redlands, CA, USA). The estimation of the SDPM and the production of the trend map were implemented in STATA version 15.0 (StataCorp, College Station, TX, USA). The results were statistically significant when two-sided *p*-values were less than 0.05.

## 3. Results

### 3.1. General Description of Mumps and Associated Variables

A total of 212,429 cases of mumps were reported in 31 provinces during 2014–2018, with an average annual reported incidence rate of 15.28/100,000, and the highest incidence rate was 18.56/100,000 in 2018. From 2014 to 2018, the average number of mumps cases in each province reached 6820, of which Hunan reported the highest number of cases at 31,443. The average temperature was 14.56 °C. The total precipitation ranged from 185.90 mm to 2939.70 mm. The five-year average of fine particulate PM2.5 in each province was 51.44 ug/m^3^. The mean values of GDP, number of registered nurses per 1000 population, urbanization rate, and proportion of the population with college-level education and above were 254.776 billion RMB, 2.57 people, 57.80% and 17.73%, respectively (Table 1).

The number of reported cases of mumps had an obvious age and seasonal distribution, with the highest peak of mumps in the 10–15-year age group and in May and June (Figure 2).

The incidence of mumps across the country is on the rise, and the incidence in Hainan exceeded 80/100,000 in 2018. The incidence rate in Hunan, Guangxi, Guizhou, Xizang, and Guangdong provinces showed a trend of fluctuating growth, while that in economically developed provinces such as Beijing, Shanghai, Zhejiang, and Jiangsu did not change and was controlled at a low level (Figure 3).

### 3.2. Spatial Distribution and 3D Trend Surface Analysis of Mumps

Incidence is divided into five categories based on the average of the maximum annual incidence rates from 2014 to 2018. The deeper the colour, the higher the incidence rate. The same classification method enables the distribution of annual mumps incidence rates to be comparable in time and space. In 2014, the incidence of mumps in Qinghai, Shanxi, Ningxia, and Chongqing was high, with an incidence between 22.34 and 33.50 per 100,000 population. In 2015, Chongqing had the highest incidence rate, exceeding 44.67/100,000. The provinces with the highest incidence in 2016 were Xinjiang, Chongqing, and Hunan. Hunan province had the highest incidence in 2017. In 2018, the incidence rate exceeded 44.67/100,000 in the Qinghai, Hunan, and Hainan provinces. Generally speaking, the provinces with high morbidity of mumps are mostly distributed in the central, southwest, and northwest regions of China, and those with low morbidity are located in the northern and coastal areas (Figure 4).

From the trend surface map, in the north-south direction, there was an obvious upward arching trend from 2014 to 2016, with the highest incidence of mumps in the central region. The incidence in the southern region was higher than that in the northern one from 2017 to 2018. In the east-west direction, the incidence in the western part was slightly higher than that in the eastern part (Figure 5).

### 3.3. Spatial Autocorrelation Analysis of Mumps

The results of global spatial autocorrelation analysis showed that Moran’s I, ranging from 0.164 to 0.389, was positive and passed the significance test for each year, which indicated spatial clustering of the disease at the provincial level in China (Table 2). The LISA map showed two types of significant clustering patterns, including high-incidence clusters (hot spots) coloured in red and low-incidence clusters (cold spots) in blue, and the location and size of these clustering regions varied slightly every year. High-High cluster areas were only located in the Gansu province in 2014 but were found in the southern provinces of China, such as Hunan, Hubei, Chongqing, Guizhou, Guangdong, and Guangxi from 2015 to 2018. Cold spots were mostly detected in the northern regions, including Heibei, Jilin, Liaoning, and Heilongjiang (Figure 6). In this study, we were more concerned with areas of High-High clustering.

### 3.4. Empirical Results

#### 3.4.1. Model Selection

In order to detect the spatial association between mumps morbidity and relevant factors, the following steps should be taken to select a suitable model. Firstly, the Lagrange multiplier (LM) test and the robust LM test is performed based on the residual of the OLS model to determine whether the model needs to include spatial terms. Then, the likelihood ratio (LR) and the Wald test are used to test whether the SDPM can be simplified to the SLPM and the SEPM. Finally, for better estimation, the Hausman test is conducted to determine whether to choose fixed effects or use random effects in the spatial regression model. The statistics of the above-mentioned tests all passed the 5% level of significance, indicating that it is feasible to choose SDPM for this study (Table 3).

#### 3.4.2. Spatial Regression Results

The goodness of fit (R2) of the time fixed effect model, individual fixed effect model, and both fixed effects model were 0.598, 0.059, and 0.009, respectively. Therefore, the optimal SDPM with time fixed effects was chosen to estimate the impact of ecological environmental factors on mumps.

The spatial correlation coefficient (ρ) in SPDM was significantly positive at the 5% level, indicating a positive spatial spillover effect (Table 4). That is to say, the prevalence rate of mumps in local areas can positively affect that in neighbouring areas. In the estimation results of the model, some variables can be seen to have a significant impact on mumps. However, in the SDPM, the coefficients of explanatory variables cannot be directly explained as in the ordinary regression model because of the existence of spatial relationships. Therefore, they were decomposed into direct effects and indirect effects using the partial differential method proposed by Lesage et al.

In our study, the direct effect of this model is the impact exerted by local explanatory variables on mumps of the local area, and the feedback effect is embraced, while the indirect effect measures the degree of influence of explanatory variables of adjacent regions on the mumps in the local region. The results of the direct effect in the SDPM revealed that the mean temperature, PM2.5, and number of registered nurses per 1000 people in the local area will have a positive impact on the incidence rate of mumps in the local region. That means that a 1% increase in the mean temperature, PM2.5, and number of registered nurses per 1000 people was associated with a 0.507%, 0.844%, and 0.962% increase in mumps, respectively. The coefficient of GDP and the proportion of the population with college-level education and above were all significantly negative, implying that the increase in the GDP and the proportion of the population with college-level education and above can clearly decrease the morbidity of mumps. The relative humidity, total precipitation, and urbanization rate did not pass the significance test at the 5% level. The indirect effect results of the model showed that the coefficient of the average temperature was significantly positive, indicating that warmer temperatures in neighbouring areas also resulted in an increase in mumps. Another significant variable is GDP: for every 1% increase of GDP in the surrounding area, the local incidence rate decreased by 0.430%. From the total effect results, the coefficients of average temperature, GDP and the proportion of the population with college-level education and above are significant, and the effect of GDP on the incidence in the country is positive (Table 5).

## 4. Discussion

As one of the most common respiratory infectious diseases, mumps has attracted widespread attention of people in China. According to a regional report by the WHO, the number of mumps cases in China accounted for 51.6% of the total global cases in 2018; therefore, the control and elimination of mumps is an urgent matter [33]. This study combined spatial autocorrelation analysis and a spatial econometric model to detect the spatial features of province-level mumps morbidity and its spatial association with ecological environmental factors. 

During the study period, it was clear that the morbidity of mumps fluctuated and increased from 13.70/100,000 in 2014 to 18.56/100,000 in 2018, consistent with a previous study [16]. This may be related to the epidemic periodicity of mumps and the attenuation of antibodies in susceptible populations. According to the time trend graph, the incidence in some provinces with high economic status remained at a relative low level. The MuCV was included in the national immunization program in 2008 in China [17], and children in most provinces were provided only one dose of the MMR vaccine, while children in economically developed provinces such as Beijing and Shanghai received two doses [34,35]. Deeks and colleagues [36] evaluated the effectiveness of the MMR vaccine during an outbreak of mumps in Canada and found that two doses of the vaccine were more effective than one. Thus, different vaccination doses and varying effectiveness of MMR vaccine between provinces may be reasons for this phenomenon. In China, the implementation of a two-dose MMR vaccination schedule for 8-month-old and 18-month-old children in June 2020 [37] is bound to decrease the national incidence of mumps and narrow the gaps between provinces to some extent. From the analysis of the spatial distribution and trend surface of mumps, we found that the prevalence rate was apparently higher in the southern and north-western areas than in the northern areas of China from 2014 to 2018. The possible reasons are as follows. First, most of the rural areas in the southern and north-western provinces are located in mountainous areas, and vaccination sites may be far from children’s homes; this, coupled with inconvenient transportation, may make it impossible to get vaccinated on time. Secondly, most of the left-behind children in rural areas are accompanied by elderly people with a lower education level, which leads to a low vaccination rate in children. Of course, the issues with vaccine management at primary health care institutions, including poor vaccine transport and storage conditions, cannot be ignored [38]. 

Regarding the bimodal seasonal and age distribution of mumps, the largest infection peaks occurred in May and June, and students under 15 years of age were the most common affected group. This finding is consistent with previous studies [12,18,39,40]. However, a study in the Netherlands showed that mumps peaked in spring and autumn, and students aged 18–25 accounted for 67.9% of cases [41]. From this point of view, the epidemiological characteristics of mumps differ between regions, but students are still susceptible. Thus, it is recommended that schools, especially boarding schools, do not ignore health education for respiratory infectious diseases and strictly implement epidemic prevention policies. At the same time, schools should give top priority to students’ physical and mental health, provide them with good accommodation and nutritious meals, and increase their levels of physical activity because waning immunity and crowded accommodation may increase the risk of contracting the virus [17,42]. In addition, we should pay attention to older adults, who are also at risk.

In this study, the results of global spatial autocorrelation analysis demonstrate that the annual reported incidence of mumps is not randomly distributed but is clustered spatially. It can be seen from the LISA cluster map that high-high clusters were mostly concentrated in the southern regions, while low-low clusters were detected in the northern regions. From a regional perspective, the southern region may be more vulnerable to social, economic, and environmental factors in neighbouring provinces [19]. From the LISA map of 2014–2018, there is indeed a phenomenon of southward movement in the high-incidence cluster areas [43], suggesting that these regions should work together to determine the cause and take effective measures.

The results of SDPM showed that mumps has a positive spatial spillover effect, which reflected obvious spatial dependence. The study found that the direct and indirect effects of the average temperature on mumps were positive, indicating that the increase of average temperature in local and neighboring provinces will increase the risk of local mumps. This finding agrees with those of previous research [12,44,45]. Contrary to the conclusion of this study, lower temperatures are more likely to affect the occurrence of mumps [13]. This discrepancy may be caused by the different research methods and study areas. In our opinion, weather conditions in neighbouring areas are not much different from local ones [19]. Therefore, it is understandable that the temperature in the surrounding provinces affects the incidence of mumps in the local provinces. This research shed light on the impacts of PM2.5 on mumps, finding a direct positive effect, while the spatial spillover effect was not significant. There are few studies on the relationship between pollutants and mumps, but a recent study also proved that PM2.5 is an extremely important pollutant that affects the occurrence of mumps [10]. PM2.5 exposure can not only make susceptible people more affected by pathogens via suppression of the immune response but also make pathogens more invasive by carrying microorganisms [8,46]. Avoiding PM2.5 exposure in susceptible populations may be effective in reducing mumps infection. Previous studies have reported that the interaction between particulate matter and high temperatures may increase the risk of pneumonia [8], but we need to explore this interaction further in mumps. It is noteworthy that the incidence of mumps was positively correlated with the number of registered nurses per 1000 people. In theory, the higher the number of registered nurses per 1000 people, which is an indicator of health resources, the lower the incidence of disease. A possible explanation is that provinces with higher nurse registrations also have higher GDP and population density, which may increase the exposure of susceptible people. It is worth mentioning that the GDP of local and neighboring provinces was negatively associated with local incidence rates of mumps. That is to say, the higher the level of economic development in the local and surrounding areas, the higher the local mumps incidence. Currently, only the study by Fu et al. has found an inverse relationship between GDP and mumps using linear regression, which is consistent with the results of our study [16]. There is no doubt that the level of GDP determines the amount of money spent on public health, which is conducive to disease control. In terms of spillover effects, inter-regional cooperation should cover all aspects, and the coordinated economic development and prevention and control of infectious diseases is not a negligible part. At present, the effect mechanism of GDP on infectious diseases is unclear, and more useful information is needed to clarify it [15]. In general, how to balance the environment, economy, and health have become a problem in need of a solution in China and the world at large.

The advantage of our study is that it is the first study to spatially analyse the association between mumps and risk factors, revealing that some factors in the surrounding area can indirectly influence the occurrence of local disease. However, our study also has several limitations. Firstly, due to the difficulty of obtaining data, this study selected data from 2014 to 2018 for analysis. Secondly, more research at the county level can be considered in future. Finally, the use of meteorological data from monitoring sites in provincial capitals may have affected the results of the study.

## 5. Conclusions

This study aimed to examine the spatial patterns of mumps and explore related influential factors using spatial analysis methods. The incidence rate displayed a clear upward trend from a temporal perspective. The areas of high-value clustering were concentrated in the southern region, while the northern region comprised low-risk areas, indicating that areas with a high incidence may be affected by factors in neighbouring areas. The higher the temperature and PM2.5, the higher the mumps incidence, the increase of GDP in local and neighboring provinces can decrease the morbidity. Our results have profound implications: firstly, in future research, we should focus on the impact of spatial factors on public health. Secondly, to achieve the common goal of reducing and eliminating diseases, mutual cooperation among regions is essential, especially in high-incidence regions. Finally, it is recommended that the government increase public health investment in areas with high incidence and balance environmental, economic, and public health issues.

## Figures and Tables

**Figure 1 ijerph-19-15429-f001:**
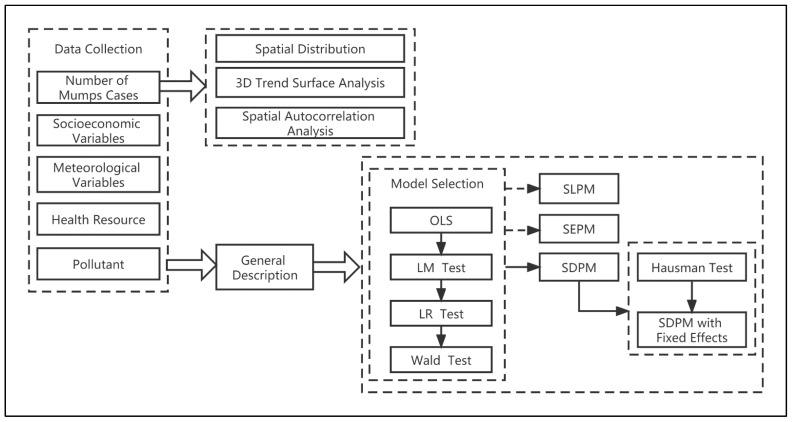
The research flow graph of this study.

**Figure 2 ijerph-19-15429-f002:**
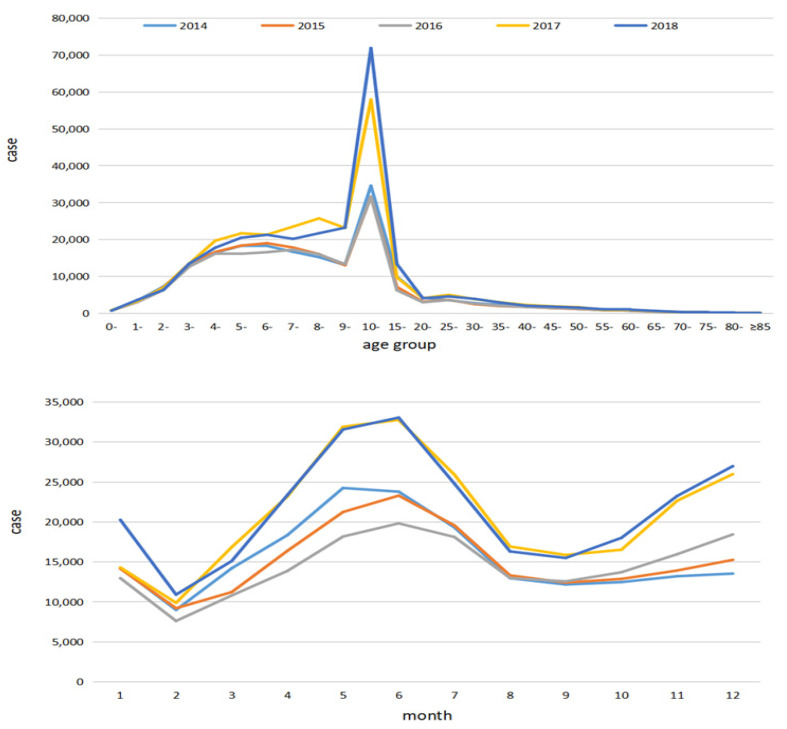
The number of reported cases of mumps in different age groups and different months in China from 2014 to 2018.

**Figure 3 ijerph-19-15429-f003:**
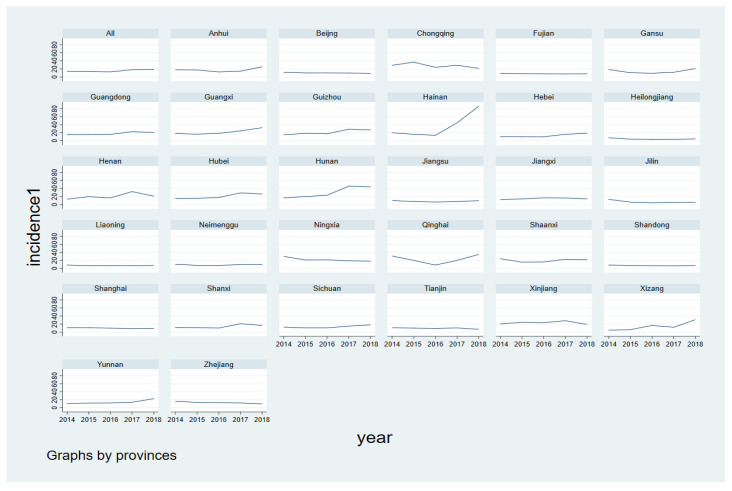
Time trend of mumps incidence in 31 provinces of China from 2014 to 2018.

**Figure 4 ijerph-19-15429-f004:**
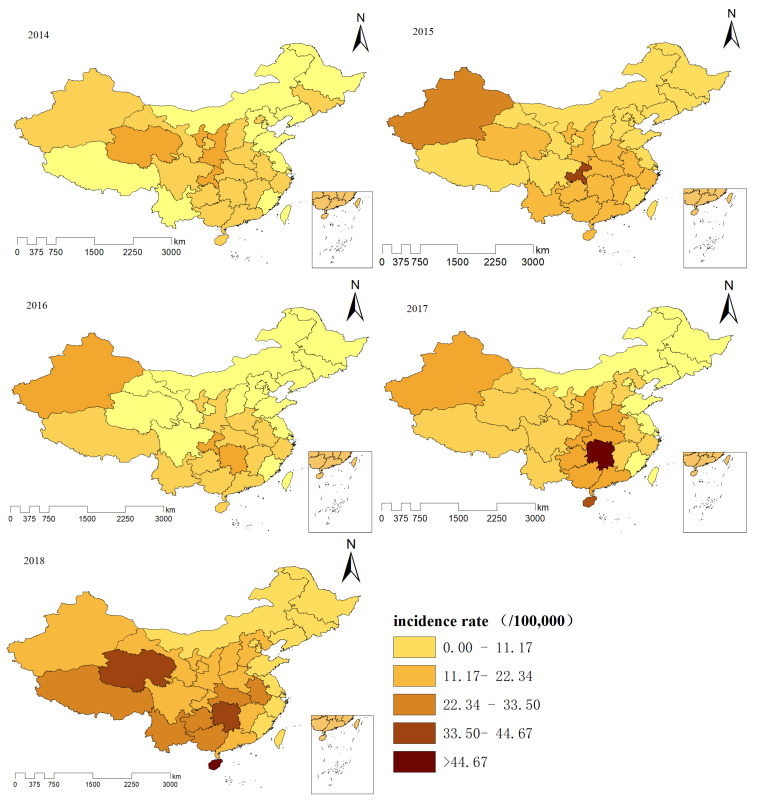
Map showing the hierarchy of the incidence rates for mumps in China from 2014 to 2018.

**Figure 5 ijerph-19-15429-f005:**
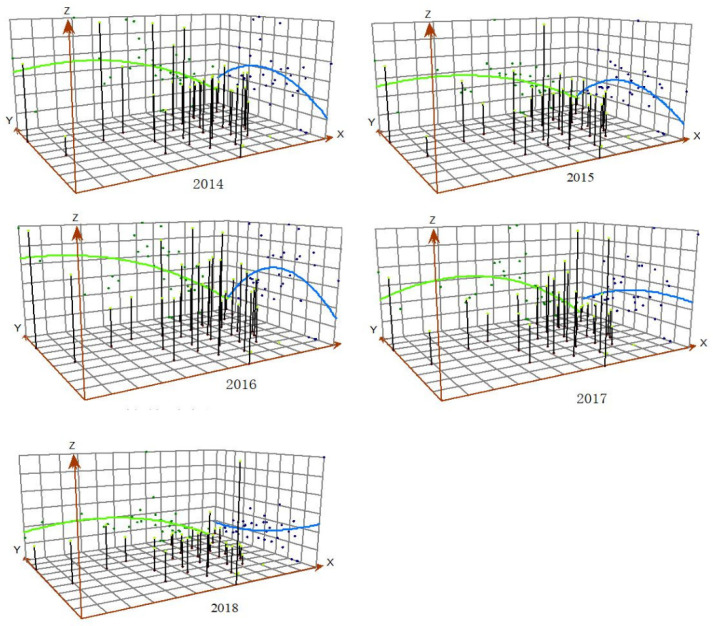
Three-dimensional trend surface analysis of mumps incidence in China from 20014 to 2018.

**Figure 6 ijerph-19-15429-f006:**
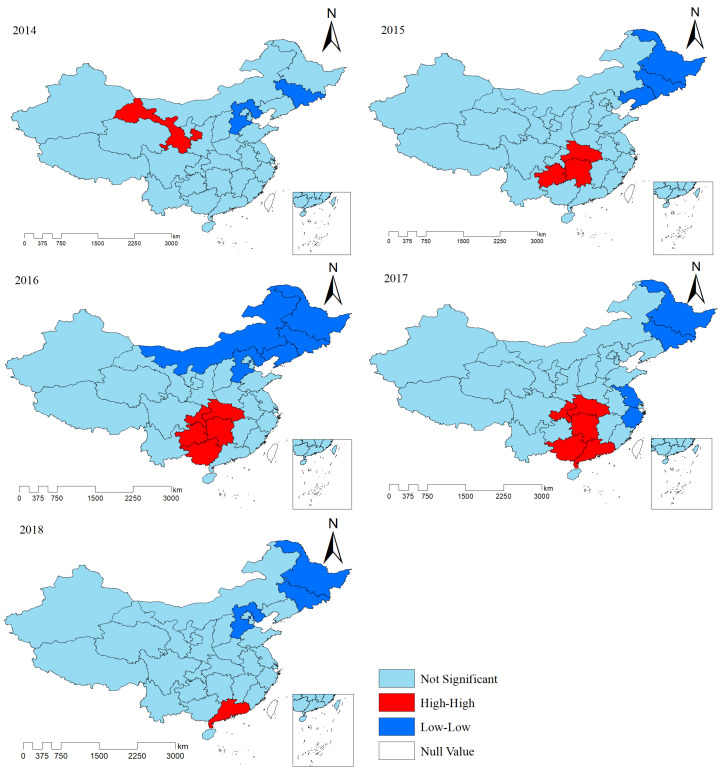
The spatial clusters of the incidence rate of mumps in China.

**Table 1 ijerph-19-15429-t001:** Descriptive statistics of the average mumps cases and ecological environment variables in all provinces from 2014 to 2018.

Variable	Observation	Mean	SD	Min	P25	P50	P75	Max
Case (person)	155	6820.29	5974.47	152.00	2265.00	5786.00	8792.00	31,443.00
TEM (°C)	155	14.56	5.05	4.95	9.53	15.61	17.63	25.33
RH (%)	155	65.79	12.25	34.33	57.25	64.08	75.67	83.75
PRE (mm)	155	992.80	585.29	185.90	480.80	880.20	1432.80	2939.70
PM2.5 (μg/m^3^)	155	51.44	18.97	12.42	37.83	49.33	63.83	118.33
GDP (/billion RMB)	155	2547.76	2034.87	92.08	1361.92	1939.96	3207.05	9727.78
RNs (per 1000)	155	2.57	0.60	0.85	2.20	2.51	2.89	4.98
UR (%)	155	57.80	12.48	25.75	50.16	56.02	64.08	89.60
EDU (%)	155	17.73	8.68	2.65	12.35	16.41	19.54	54.78

**Table 2 ijerph-19-15429-t002:** Global autocorrelation analysis of the incidence of mumps in China from 2014 to 2018.

Variables	*I*	z	*p*-Value
Overall	0.30	5.75	<0.001
2014	0.16	1.70	0.044
2015	0.22	2.24	0.013
2016	0.39	3.53	<0.001
2017	0.39	3.64	<0.001
2018	0.20	2.37	0.009

**Table 3 ijerph-19-15429-t003:** Model test results.

Test	Statistic	*p*-Value
LM error test	4.62	0.032
Robust LM error test	10.52	0.001
LM lag test	22.36	<0.001
Robust LM lag test	28.27	<0.001
Wald lag test	29.15	<0.001
Wald error test	27.99	<0.001
LR test (SDM & SLM)	26.79	<0.001
LR test (SDM & SEM)	25.56	0.001
Hausman test	28.51	<0.001

**Table 4 ijerph-19-15429-t004:** The regression results of the Spatial Durbin Panel Model.

Variables	Time-Fixed Effects	Ind-Fixed Effects	Both-Fixed Effects
lnTEM	0.46 ** (0.18)	−0.77 (0.95)	0.57 (0.93)
lnPM2.5	0.86 *** (0.15)	0.49 ** (0.23)	0.50 ** (0.23)
RH	0.00 (0.01)	0.01 (0.01)	0.024 (0.01)
lnPRE	0.14 (0.16)	0.06 (0.14)	0.14 (0.13)
lnGDP	−0.43 *** (0.06)	−0.02 (0.44)	−0.08 (0.43)
lnRNs	0.91 *** (0.27)	2.12 *** (0.58)	2.11 *** (0.57)
UR	0.00 (0.01)	0.06 * (0.03)	0.06 (0.03)
EDU	−0.01 * (0.01)	0.03 ** (0.01)	0.03 ** (0.01)
W.lnTEM	0.61 * (0.36)	−2.67 * (1.63)	0.41 (1.83)
W.lnPM2.5	−0.39 (0.34)	−0.29 (0.37)	−0.24 (0.44)
W.RH	0.02 ** (0.01)	0.02 (0.02)	0.06 ** (0.03)
W.lnPRE	−0.45 * (0.23)	−0.60 *** (0.20)	−0.29 (0.22)
W.lnGDP	−0.21 * (0.15)	1.03 (0.67)	0.90 * (0.68)
W.lnRNs	0.66 * (0.58)	−1.05 (1.03)	−0.85 (1.32)
W.UR	−0.00 (0.01)	−0.09 (0.07)	−0.01 (0.07)
W.EDU	−0.00 (0.02)	−0.04 ** (0.02)	−0.01 (0.03)
ρ	0.02	0.01	0.34
R2	0.60	0.06	0.01
Log-likelihood	−40.44	13.19	21.03

Note: *, **, and *** represent significance at the 10, 5, and 1% levels, respectively.

**Table 5 ijerph-19-15429-t005:** Spatial effect decomposition results of the Spatial Durbin Panel Model.

Variables	Direct Effect	Indirect Effects	Total Effects
lnTEM	0.51 *** (0.18)	0.95 ** (0.42)	1.46 *** (0.36)
lnPM2.5	0.84 *** (0.14)	−0.20 (0.41)	0.65 (0.42)
RH	0.00 (0.01)	0.03 * (0.02)	0.03 * (0.02)
lnPRE	0.10 (0.15)	−0.52 * (0.31)	−0.42 (0.34)
lnGDP	−0.45 *** (0.06)	−0.43 ** (0.20)	−0.88 *** (0.23)
lnRNs	0.96 *** (0.28)	1.18 (0.79)	2.14 ** (1.00)
UR	−0.00 (0.01)	−0.00 (0.02)	−0.00 (0.02)
EDU	−0.02 * (0.01)	−0.01 (0.03)	−0.03 (0.03)

Note: *, **, and *** represent significance at the 10, 5, and 1% levels, respectively.

## Data Availability

Data on mumps cases in 31 administrative regions of China from January 2014 to December 2018 were collected from the Data Center of China Public Health Science (https://www.phsciencedata.cn/Share/ky_sjml.jsp, accessed on 25 July 2022) [23]. Meteorological and socio-economic data were obtained from the National Bureau of Statistics (http://www.stats.gov.cn/tjsj/ndsj/, accessed on 25 July 2022) [24]. PM2.5 data were extracted from the Air Quality Online Monitoring and Analysis Platform (https://www.aqistudy.cn/historydata/, accessed on 25 July 2022) [25].

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
