# Peer review of "Spatial Effect of Ecological Environmental Factors on Mumps in China during 2014–2018"

_ijerph, 2022, doi:10.3390/ijerph192315429_

Round 1
Reviewer 1 Report
Dear authors,
I think that the references used from lines 98 to 112 in the materials and methods section should be more adequately cited, leading information, such as internet address and data access date, to the specific references section.
I was doubtful about the statement in lines 411 and 412, in the conclusions section, in which it is stated that GDP can positively affect morbidity. Was it not found that the GDP presented an inverse association with the occurrence of the outcome?
Author Response
Response to Reviewer 1 Comments
Point 1: I think that the references used from lines 98 to 112 in the materials and methods section should be more adequately cited, leading information, such as internet address and data access date, to the specific references section.
Response 1: Thank you very much for your suggestions, and we have added the data source information (Internet address and data acquisition time) in the Materials and Methods section to the references. We put the references in lines 488-494.
Point 2: I was doubtful about the statement in lines 411 and 412, in the conclusions section, in which it is stated that GDP can positively affect morbidity. Was it not found that the GDP presented an inverse association with the occurrence of the outcome?
Response 2: Thank you very much for your questions about the statements on lines 411 and 412 in the conclusions section of this study. After careful inspection again, we did find that there were problems in the expression of lines 411 and 412, and we have revised it. The modified position is in lines 408-409. On the question of whether GDP is found to be negatively correlated with the occurrence of outcome, the results of the spatial Durbin panel model (SDPM) show that the increase of GDP in local and neighboring areas can reduce the incidence, so the GDP presented an inverse association with the incidence. This may be a misunderstanding caused by our unclear language expression.

Reviewer 2 Report
The authors provided us with a detailed and complete analysis with a huge amount of data. They analyzed the impact of Geographical factors, meteorological conditions, and economics on disease incidence. This study provides information on the effects of temperature, and economic levels on the transmissions in different regions which could facilitate the prevention of mumps occurrence. In a word, the overall structure of the article is complete, but some issues need to be addressed:
1. It is helpful if the author could provide a complete flow chart of the study.
2. Is there an English version of these data sources used in this study, eg, “Data Center of China Public Health Science established by the Chinese Center 97 for Disease Control and Prevention (CDC) (https://www.phscience-98 data.cn/Share/ky_sjml.jsp”?
3. Careful proofreading and polishing are required before its publication.
4. line 229, it should be “2014 to 2018“.
5. The discussion section might be shortened for better reading. Could the authors provide explanations on why there was an evident upward trend from 2014 to 2018?
Author Response
Response to Reviewer 2 Comments
Point 1: It is helpful if the author could provide a complete flow chart of the study.
Response 1: We gratefully appreciate for your valuable suggestion. According to your comments, we have added the flow chart of this study (Figure 1)
Point 2:Is there an English version of these data sources used in this study, eg, “Data Center of China Public Health Science established by the Chinese Center 97 for Disease Control and Prevention (CDC) (https://www.phscience-98 data.cn/Share/ky_sjml.jsp”?
Response 2: Thank you! At present, there is no English version of these data sources used in this study.
Point 3: Careful proofreading and polishing are required before its publication.
Response 3: Thank you for your comments, our article has been carefully proofread and polished by the professional team.
Point 4: line 229, it should be “2014 to 2018”.
Response 4: Thank you for pointing out the writing error in line 229 of the article, which is caused by our carelessly. We have corrected this error and have also carefully checked the article.
Point 5: The discussion section might be shortened for better reading. Could the authors provide explanations on why there was an evident upward trend from 2014 to 2018?
Response 5: Thank you very much for your valuable comments on this study. In order to facilitate readers' reading, we shortened the length of discussion section by modifying some contents. In response to your question about why there was an evident upward trend in incidence from 2014 to 2018, the epidemic cycle with an interval of 3-5 years and the attenuation of antibodies in the population may explain this phenomenon.We also add the explanation to lines 314 and 315 in the discussion section of this study.
